# Video Diffusion Models Learn the Structure of the Dynamic World

## Abstract

Diffusion models have demonstrated significant progress in visual perception tasks due to their ability to capture fine-grained, object-centric features through large-scale vision-language pretraining. While their success in image-based tasks is well-established, extending this capability to the domain of video understanding remains a key challenge. In this work, we explore the potential of diffusion models for video understanding by analyzing the feature representations learned by both image- and video-based diffusion models, alongside non-generative, self-supervised approaches. We propose a unified probing framework to evaluate six models across four core video understanding tasks: action recognition, object discovery, scene understanding, and label propagation. Our findings reveal that video diffusion models consistently rank among the top performers, particularly excelling at modeling temporal dynamics and scene structure. This observation not only sets them apart from image-based diffusion models but also opens a new direction for advancing video understanding, offering a fresh alternative to traditional discriminative pre-training objectives. Interestingly, we demonstrate that higher generation performance does not always correlate with improved performance in downstream tasks, highlighting the importance of careful representation selection. Overall, our results suggest that video diffusion models hold substantial promise for video understanding by effectively capturing both spatial and temporal information, positioning them as strong competitors in this evolving domain.

## 1 Introduction

Beyond generating high-fidelity images, diffusion models have achieved significant breakthroughs in visual perception. Their success is largely attributed to the large-scale vision-language pretaining, which allows them to capture detailed, object-centric features. This has positioned them as strong candidates for tasks such as image segmentation (Zhao et al., 2023; Xu et al., 2023) and classification (Li et al., 2023). This raises a natural question: *Can diffusion models' success in images extend to the more complex domain of video understanding?*

Video understanding presents unique challenges absent in the image domain, particularly in capturing *temporal dynamics and motion patterns*. While image diffusion models have demonstrated success in video-level tasks like mask propagation (Tang et al., 2023), they do not explicitly model time and thus struggle with higher-level video understanding. In contrast, video diffusion models (Blattmann et al., 2023a; Wang et al., 2023b) are inherently designed to capture spatial-temporal dynamics, making them far better suited for these tasks. As illustrated in Figure 1, where we visualize video representations using K-Means clustering and three-channel PCA for several widely used visual foundation models, video diffusion models excel at capturing motion dynamics - a critical capability that sets them apart from their image-based counterparts. Additionally, video diffusion models retain a high-level structured representation of the video input, further enhancing their position as strong contenders for advanced video understanding tasks.

To investigate this advantage in more depth, we propose a unified probing framework to analyze feature representations across a range of visual models. Our study spans six models, including both image- and video-based architectures, as well as non-diffusion (Oquab et al., 2023; Bardes et al., 2024a) and diffusion-based approaches. In the diffusion category, we further evaluate both UNet-based (Blattmann et al., 2023a; Rombach et al., 2022; Wang et al., 2023b) and diffusion-

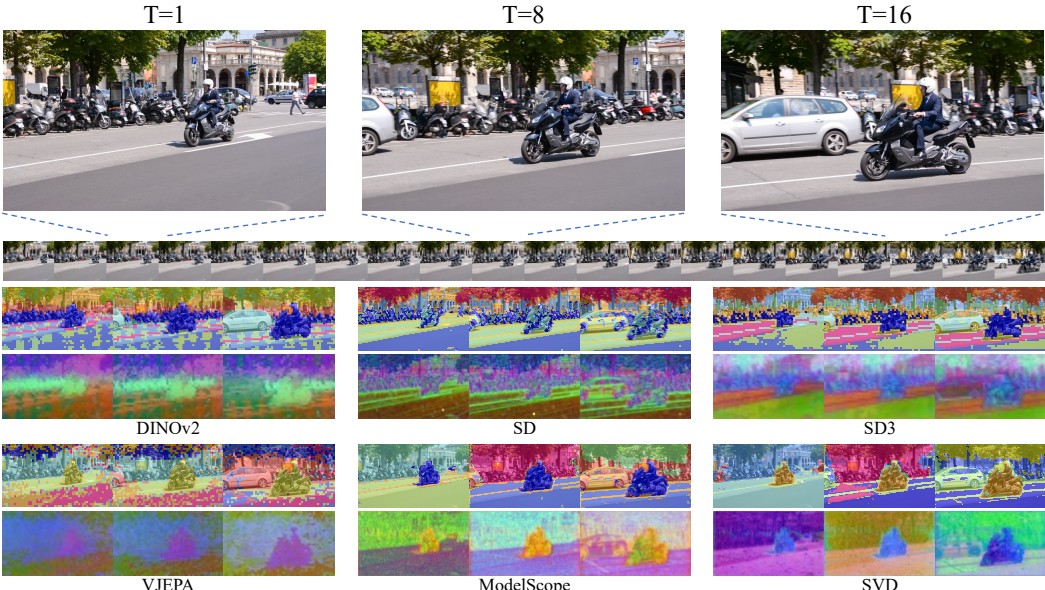

Figure 1: Video feature visualizations on DAVIS17 (Pont-Tuset et al., 2017) dataset. Row 1: K-Means clusters (K=10); Row 2: three-channel PCA visualizations. Compared to image diffusion models, video diffusion models excel at capturing motion dynamics while retaining a higher-level structured representation of the video input compared to conventional models. These unique characteristics position them as strong candidates for advanced video understanding.

transformer-based techniques (Esser et al., 2024; Zheng et al., 2024; Peebles & Xie, 2023). Our evaluation focuses on four key tasks that highlight different aspects of video understanding: (1) *action recognition*, a supervised classification task for assessing global video-level representations; (2) *object discovery*, an unsupervised segmentation task measuring dense feature quality; (3) *scene understanding*, a supervised task to test the semantic and geometrical awareness of dense feature maps; and (4) *label propagation*, a training-free task evaluating the temporal consistency of features. These tasks collectively provide a comprehensive examination of the strengths and weaknesses of each model across various facets of video understanding.

Our evaluation reveals that video diffusion models excel at **capturing the structure of the dynamic world**, making them consistently rank among the top performers across different tasks in video understanding. Key insights from our findings include:

- **Motion and temporal dynamics**: Video diffusion models demonstrate exceptional proficiency in capturing motion patterns and temporal dynamics, a capability that significantly contributes to their strong performance in video understanding tasks.

- **Comparison with image diffusion models**: While image diffusion models learn object-centric and semantic-aware representations, video diffusion models retain a high-level structured representation while effectively modeling spatio-temporal information. This enables them to generally outperform their image-based counterparts, particularly in tasks that rely on training.

- **Role of training data**: The scale and nature of the training data play a key role in model performance. Models trained on larger datasets exhibit greater robustness, and video pre-training enhances motion modeling capabilities at the potential cost of a loss in temporal consistency when handling static objects.

- **Generation v.s. perception**: Interestingly, in diffusion models, a higher capacity for generation does not always correlate with improved performance in visual perception tasks. Different versions of the same model can be optimal for different downstream tasks, and no universal metric for selecting a representation exists as of yet.

Overall, our results suggest that video diffusion models hold substantial promise for video understanding by effectively capturing both spatial and temporal information, positioning them as strong competitors in this evolving domain.

## 2 RELATED WORK

**Diffusion models**, inspired by principles of heat and anisotropic diffusion, have emerged as a powerful class of generative models for image and video synthesis (Perona & Malik, 1990; Weickert et al., 1998). Recent advancements have positioned diffusion models as state-of-the-art across unconditional (Ho et al., 2020; Song et al., 2020a;b; Dhariwal & Nichol, 2021; Bond-Taylor et al., 2022) and conditional image synthesis tasks (Nichol et al., 2021; Rombach et al., 2022; Saharia et al., 2022; Ramesh et al., 2022; Gu et al., 2022; Yu et al., 2022; Ho & Salimans, 2022; Wang et al., 2022; Zhang & Agrawala, 2023). Notably, Denoising Diffusion Probabilistic Models (DDPMs) (Ho et al., 2020) introduced the use of neural networks for modeling the denoising process, optimizing with a weighted variational bound. The Denoising Diffusion Implicit Model (DDIM) (Ho et al., 2020) enhanced this by incorporating a non-Markov sampling strategy to accelerate inference. Stable Diffusion (Rombach et al., 2022) extended the diffusion-denoising process into the latent space of a pre-trained autoencoder (Kingma & Welling, 2013), enabling more efficient large-scale model training. More recently, Transformer-based models have been introduced to further scale up training, achieving superior performance (Peebles & Xie, 2023; Esser et al., 2024).

The extension of diffusion models from image to video generation (Ho et al., 2022b; He et al., 2022; Luo et al., 2023b) represents a significant advancement, encompassing both text-to-video (T2V)(Blattmann et al., 2023b; Qing et al., 2023; Khachatryan et al., 2023; Wang et al., 2023c; Jain et al., 2023) and image-to-video (I2V) generation(Guo et al., 2023a; Wang et al., 2023a; Zhang et al., 2023b; Ni et al., 2023).These efforts largely build upon pre-trained image-level diffusion models, such as Stable Diffusion (Rombach et al., 2022), by training the additional video backbone with extra video data (Ho et al., 2022a; Wang et al., 2023b; Guo et al., 2023b; Chen et al., 2023a; Girdhar et al., 2023; Blattmann et al., 2023a; Chen et al., 2024). Some approaches avoid retraining entirely by utilizing training-free algorithms for video generation from image models (Wu et al., 2023; Singer et al., 2022; Yuan et al., 2023) or employing training-free algorithms for direct video generation from image models (Wu et al., 2023; Singer et al., 2022; Yuan et al., 2023). Most recently, Sora (Brooks et al., 2024) and its open-sourced couterparts (Zheng et al., 2024; Lab & etc., 2024) demonstrated leading video generation capabilities with the more advanced architecture of diffusion transformer (Peebles & Xie, 2023). Among them, ModelscopeT2V (Wang et al., 2023b), Stable Video Diffusion (SVD) (Blattmann et al., 2023a), and OpenSora (Zhang et al., 2023b) have open-sourced their large-scale pre-trained model which serves as our backbones for this study.

**Diffusion models for visual perception.** Diffusion models have also demonstrated strong semantic correspondence in their feature spaces (Hertz et al., 2022; Tang et al., 2023; Zhang et al., 2023a). This has spurred a line of research that utilizes diffusion models for visual perceptual tasks, through either training diffusion-based models for specific tasks such as segmentation (Xu et al., 2023; Zhao et al., 2023; Ozguroglu et al., 2024), depth estimation (Saxena et al., 2023b;a; Guizilini et al., 2024) or open-world novel view synthesis (Liu et al., 2023). ther work leverages pre-trained *frozen* diffusion models for perceptual learning (Tang et al., 2023; Luo et al., 2023a; Zhang et al., 2023a; Hedlin et al., 2023; Namekata et al., 2024; Khani et al., 2023), or explores their use in data augmentation for discriminative tasks (Trabucco et al., 2024; Feng et al., 2023; Burg et al., 2023; Meng et al., 2021).

Among them, DIFT (Tang et al., 2023) proposes a general pipeline to extract features from real images with diffusion models, for which we adopt for our evaluation pipeline. Chen et al. (2023b) and Nag et al. (2023) leverage diffusion models for video-related tasks, but they *do not* leverage a video diffusion model with spatial-temporal reasoning modules. VD-IT (Zhu et al., 2024) designs a novel architecture with video diffusion models as the backbone for referring object segmentation. Lexicon3D (Man et al., 2024) conducted a comprehensive study of visual foundation models, including diffusion-based ones, on 3D scene understanding. Unlike previous work, this study addresses the general video understanding with diffusion models across multiple tasks, each with a distinct focus.

## 3 PROBING VIDEO UNDERSTANDING WITH DIFFUSION MODELS

### 3.1 PRELIMINARY: DIFFUSION MODELS

**Latent diffusion models.** Diffusion models (Ho et al., 2020) are latent variable models that learn the data distribution with the inverse of a Markov noising process. Latent diffusion models

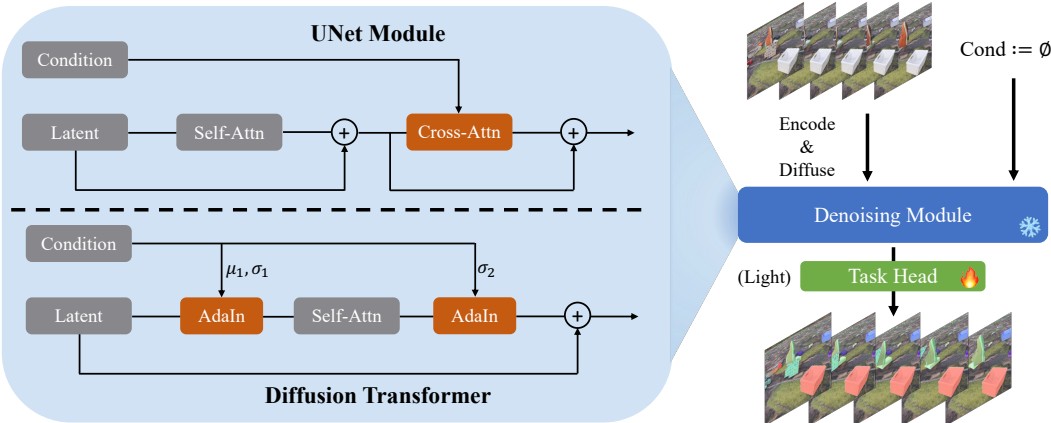

Figure 2: The architecture of our probing framework for video understanding using diffusion models. Video feature representations are extracted from the denoising module, followed by a lightweight task head to produce task-specific annotations. The process of feature extraction from UNet or DiT models is illustrated on the left.

(LDM) (Rombach et al., 2022) further switch the diffusion-denoising mechanism from RGB space to latent space, which improves the scalability and enables large-scale training. Concretely, an encoder $\mathcal{E}$ is trained to map a given image $x \in \mathcal{X}$ into a spatial latent code $z = \mathcal{E}(x)$. A decoder $\mathcal{D}$ is then tasked with reconstructing the input image such that $\mathcal{D}(\mathcal{E}(x)) \approx x$.

Considering the clean latent $z_0 \sim q(z_0)$, where $q(z_0)$ is the posterior distribution of $z_0$, LDM gradually adds Gaussian noise to $z_0$ in the *diffusion process*:

$$q(z_t|z_{t-1}) = \mathcal{N}(z_t; \sqrt{1-\beta_t}z_{t-1}, \beta_t\mathbf{I}), \tag{1}$$

where $\beta_t$ is a variance schedule that controls the strength of the noise added in each timestep. We can derive a closed-form process from Equation 1 to convert a clean latent $z_0$ to a noisy latent $z_T$ of arbitrary timestep $T$:

$$z_T \sim q(z_T|z_0) = \mathcal{N}(z_T; \sqrt{\bar{\alpha}_T}z_0, (1-\bar{\alpha}_T)\mathbf{I}), \tag{2}$$

where the notations $\alpha_T = 1 - \beta_T$ and $\bar{\alpha}_T = \prod_{s=1}^{T} \alpha_s$ make the formulation concise. When $T \to \infty$, $z_T$ is nearly equivalent to sampling from an isotropic Gaussian distribution.

The denoising process takes inverse operations from the diffusion. We estimate the denoised latent at timestep $t-1$ from $t$ by:

$$p_\theta(z_{t-1}|z_t) = \mathcal{N}(z_{t-1}; \mu_\theta(z_t, t), \mathbf{\Sigma}_\theta(z_t, t)), \tag{3}$$

where the parameters $\mu_\theta(z_t, t), \mathbf{\Sigma}_\theta(z_t, t)$ of the Gaussian distribution can be estimated from the model. As revealed by Ho et al. (2020), $\mathbf{\Sigma}_\theta(z_t, t)$ has few affects on the results experimentally, therefore estimating $\mu_\theta(z_t, t)$ becomes the main objective. A reparameterization is introduced to estimate it:

$$\mu_\theta(z_t, t) = \frac{1}{\sqrt{\alpha_t}}\left(z_t - \frac{\beta_t}{\sqrt{1-\bar{\alpha}_t}}\epsilon_\theta(z_t, t)\right), \tag{4}$$

where $\epsilon_\theta(z_t, t)$ is typically a denoising UNet module (Ronneberger et al., 2015) or diffusion transformer (Peebles & Xie, 2023) module. $\epsilon_\theta(z_t, t)$ is usually conditioned on additional inputs, such as texts or image embeddings, to steer the denoising trajectory. In Figure 2 (left), we demonstrate how the extra modality is fused to the latent space: for UNet-based models, cross-attention modules are utilized to fuse the features while for DiT-based models, the additional embedding is fused via AdaIn (Huang & Belongie, 2017) modules or broadcasted self-attention. The final objective of latent diffusion models is:

$$\mathcal{L}_{\text{LDM}} := \mathbb{E}_{\mathcal{E}(x), \epsilon \sim \mathcal{N}(0,1), t}\left[\|\epsilon - \epsilon_\theta(z_t, t)\|_2^2\right]. \tag{5}$$

**Video diffusion models.** Video diffusion models generally share a similar architecture to the 2D diffusion models. Given a video $\mathbf{v} = [x^1, x^2, \cdots, x^N]$, a spatial encoder $\mathcal{E}^v$ is applied to each frame

to map them to the latent code $z^i = \mathcal{E}^v(x^i)$, where $i$ is the frame index. We use the annotation $\mathbf{z} = [z^1, z^2, \cdots, z^N]$ for convenience. For the decoder, usually, a temporal-spatial decoder is applied to enforce the temporal consistency $\mathcal{D}^v(\mathbf{z}) \approx \mathbf{v}$.

One crucial distinction for video diffusion models is: the inductive noise is applied at the video level instead of the frame level. To accommodate this, the denoising UNet, denoted as $\epsilon_\theta^v$, has been redesigned to 3D by either introducing additional temporal attention (Vaswani et al., 2017) modules (Blattmann et al., 2023a), or replacing the spatial attention modules with spatial-temporal attention modules (Zhang et al., 2023b; Wang et al., 2023b).

### 3.2 Video Understanding Probing Framework

We show our unified probing framework in Figure 2. We fetch the video representation from the denoising module and apply a lightweight task head afterward for different tasks.

**Diffusion features.** We extract video features with diffusion models following DIFT (Tang et al., 2023). The process begins by adding noise at timestep T to the real video latent (Equation 2), moving it into the $\mathbf{z}T$ distribution. This noisy video latent, along with T, is then passed to $\epsilon^v\theta$. Instead of using the final output of $\epsilon_\theta^v$, which predicts the noise, we extract features from intermediate layer activations that effectively capture the video's underlying representations. These intermediate representations form the diffusion features. For features from image diffusion models, we follow a nearly identical process, except that we handle the videos frame by frame. Additionally, during feature extraction, we introduce a fixed "null-embedding" as the condition for $\epsilon_\theta^v$. For language-based models, this embedding is obtained by passing an empty prompt to the text encoder. For image-based models, we use an all-zero conditional image.

**Adaptation for downstream tasks.** After extracting features from diffusion models, we use a lightweight task head (much fewer than 1% of the backbone's parameters) to interpret these features and generate annotations for the specified tasks, as demonstrated by the object discovery task in Figure 2. It is important to note that the diffusion backbone remains frozen during our evaluations unless otherwise specified. In addition, our study also includes non-diffusion feature extractors, where we simply replace the diffusion features with representations from other visual models.

## 4 Experimental Evaluations

### 4.1 Evaluation Settings

**Baseline models.** We perform our video understanding analysis with six visual foundation models. DINOv2 (Oquab et al., 2023) is a self-supervised approach that employs a conservative learning algorithm, leveraging both image-level and patch-level similarity matching to produce robust visual features. VJEPA (Bardes et al., 2024b) learns comprehensive video representations by reconstructing from masked video patch features, enabling it to capture spatiotemporal dynamics. Stable Diffusion (SD)(Rombach et al., 2022) is a text-to-image diffusion model operating in the latent space. Through large-scale vision-language generative pretraining, it learns rich object-centric representations suitable for various perception tasks. Stable Diffusion 3 (SD3)(Esser et al., 2024) utilizes a DiT-based (Peebles & Xie, 2023) architecture to fuse multi-modal embeddings and incorporates rectified flow for more efficient training, therefore yielding enhanced performance compared with SD. Building upon SD, ModelScopeT2V (Wang et al., 2023b) further adds temporal modeling units to produce promising results in text-to-video generation. Stable Video Diffusion (SVD) is an image-to-video generation model that trains its temporal-spatial modules on a large-scale dataset, excelling at capturing object motion during video generation. Detailed configurations of these feature extractors are provided in Table 1.

**Tasks.** Our evaluation focuses on four diverse tasks, each targeting a different dimension of video understanding: (1) **Action recognition** is a video classification task aimed at identifying the actions taking place in a given video. This task is primarily used to evaluate how well models can capture global, video-level representations; (2) **Object discovery** is a self-supervised instance segmentation task that focuses on identifying and tracking dynamic objects in videos. It evaluates the model's ability to extract dense, fine-grained features necessary for distinguishing object-centric details; (3)

| Model | Type | Architecture | Dataset (Scale) | Feature Dim | Downsample Ratio |
|---|---|---|---|---|---|
| DINOv2 (Oquab et al., 2023) | Image | ViT-L | LVD-142M (142M) | 1024 | 14 |
| VJEPA (Bardes et al., 2024b) | Video | ViT-L | VideoMix2M (2M) | 1024 | 16 |
| SD (Rombach et al., 2022) | Image | UNet | LAION (5B) | 1280/640 | 8/16 |
| SD3 (Esser et al., 2024) | Image | DiT | Public Images (1B) | 1536 | 16 |
| ModelScope (Wang et al., 2023b) | Video | UNet | WebVid (10M) | 1280/640 | 8/16 |
| SVD (Blattmann et al., 2023a) | Video | UNet | LVD (152M) | 1280/640 | 8/16 |

Table 1: Details of the pretrained feature extractors we used for our video understanding evaluation.

**Scene understanding** involves interpreting the semantic and geometric aspects of the video. We use video semantic segmentation and monocular depth estimation as examples, allowing us to test the model's understanding of both scene content and structure; and (4) **Label propagation** involves propagating annotations, usually instance masks or key points, from the first frame across the entire video. This training-free task assesses how well models maintain temporal consistency by matching feature similarities across frames. Together, these tasks offer a comprehensive view of the models' strengths and weaknesses across multiple facets of video understanding.

**Datasets and metrics.** We evaluate *action recognition* recognition with top 1 and top 5 accuracy on UCF101 (Soomro et al., 2012) and HMDB51 (Kuehne et al., 2011). We study the object discovery task on MOVi-C and MOVi-E (Greff et al., 2022), and take foreground adjust random index (FG. ARI) and video mean best overlap (mBO) as metrics. For object tracking, we evaluate on MOT17 (), and report xxx. We conduct the label propagation for video object segmentation on DAVIS17 (Pont-Tuset et al., 2017) and keypoint estimation on JHMDB (Jhuang et al., 2013) following the same setup as DIFT (Tang et al., 2023). We report region-based similarity $\mathcal{J}$ and contour-based accuracy $\mathcal{F}$ (Perazzi et al., 2016) for DAVIS17, and percentage of correct keypoints (PCK) for JHMDB.

**Key implementation details.** For action recognition, we apply a single-layer MLP on top of the averaged features among all the patches. For object discovery and object tracking, we build upon MoTok (Bao et al., 2023) and trackformer (Meinhardt et al., 2022) by replacing their encoders with the evaluated feature extractors. With the inspiration that earlier layers of diffusion models take higher-level representation while later layers contain more object-level ones, we design the use of block index 1 (for SD, ModelScope, and SVD) and block index 12 (for SD3 and Open-Sora) for action recognition. For the other tasks, we use block index 2 and block index 24 respectively. We use the noise level 50 by default, with a corresponding timestep T=50 (for SD, ModelScope, and SVD) or T=16 (for SD3 and Open-Sora). We use batch size 12 with 4 NVIDIA-A100 GPUs running in parallel for all the backbones except ModelScope. We use batch size 6 with 8 GPUs in parallel for it to fit its CUDA requirement.

More details about datasets, model implementation, and training configurations are included in Section A in the Appendix.

## 4.2 MAIN RESULTS

In this section, we show the results for the four tasks individually. Visualizations for top performers, DINOv2 (Oquab et al., 2023), SD (Rombach et al., 2022), ModelScope (Wang et al., 2023b), and SVD (Blattmann et al., 2023a) are shown in Figure 3. Besides the mentioned six visual foundation models, we additionally include a DiT-based video diffusion model, Open-Sora (Zheng et al., 2024), into the evaluation of action recognition. We do not include this model for other evaluations since it fails to produce precise patch-wise presentations. We first show the task-dependent observations followed by a few overall conclusions based on the whole set of experiments.

**Action recognition.** With the results shown in Table 2 (left), we draw the following observations: (1) Surprisingly, we find that SVD achieves the best performance on UCF-101 and second best on JHMDB, consistently outperforming image diffusion models, and the conventional DINOv2 encoder. These results indicate that a powerful video diffusion model has the ability to capture the global-level information of a video. (2) For diffusion-based models, the training data plays a key role in the performance. Compared to SD3, SD is trained on a larger-scale image set, yielding a better performance. The same conclusion holds for the comparison between SVD, ModelScope, and Open-Sora. (3) Current diffusion-based feature extraction pipelines with UNet-based models

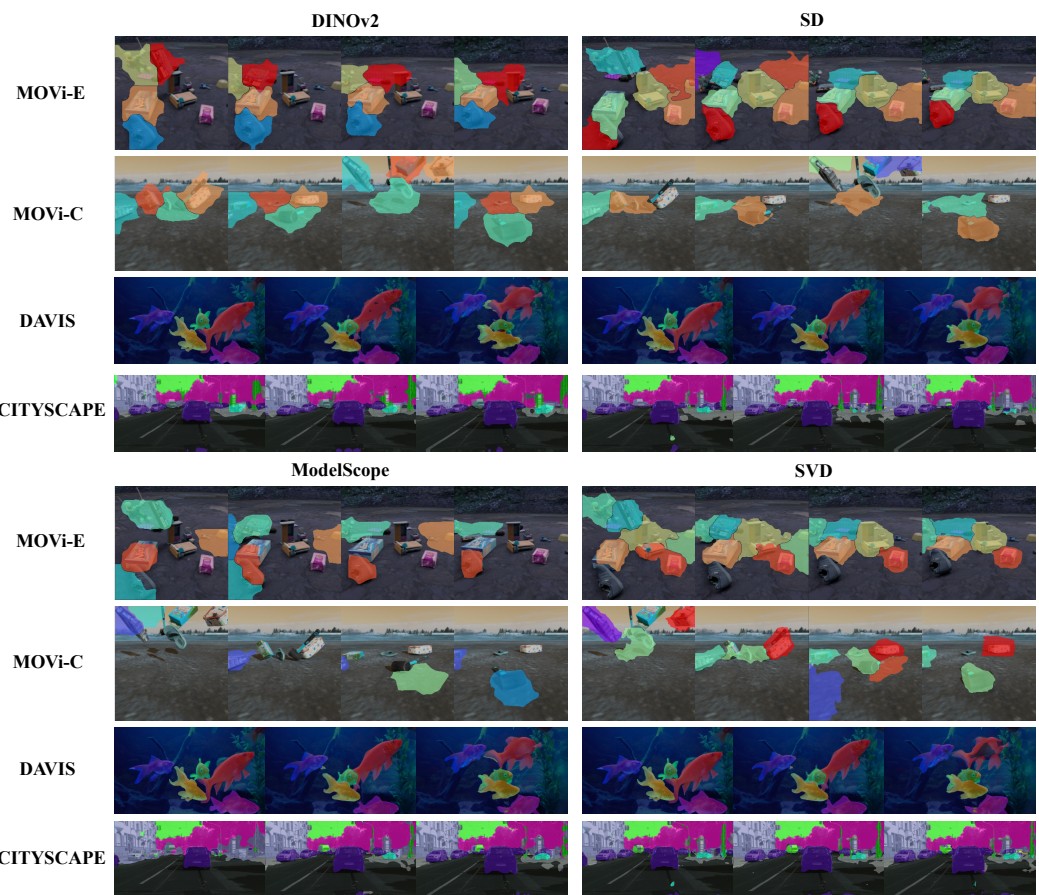

Figure 3: Visual comparisons for top performers, DINOv2 (Oquab et al., 2023), SD (Rombach et al., 2022), ModelScope (Wang et al., 2023b), and SVD (Blattmann et al., 2023a) regarding object discovery, label propagation, and scene understanding. DINOv2 works better for semantic understanding while diffusion-based models provide excel at object-centric tasks.

| Backbone | UCF101 | | HMDB51 | | MOVi-C | | MOVi-E | |
|---|---|---|---|---|---|---|---|---|
| | Top1 Acc | Top 5 Acc | Top1 Acc | Top 5 Acc | FG.ARI | mBO | FG.ARI | mBO |
| DINOv2 | 89.8 | 97.8 | 61.6 | 89.6 | 55.6 | 29.2 | 71.9 | 26.3 |
| VJEPA | 92.1 | 98.5 | 66.5 | 92.3 | 31.8 | 18.6 | 49.9 | 18.0 |
| SD | 63.5 | 86.1 | 33.0 | 68.1 | 40.6 | 24.8 | 63.4 | 26.9 |
| SD3 | 60.9 | 85.8 | 32.4 | 62.1 | 43.3 | 26.3 | 65.1 | 28.6 |
| ModelScope | 80.6 | 94.9 | 50.7 | 80.2 | 41.3 | 25.1 | 63.7 | 27.5 |
| SVD | 92.3 | 98.6 | 63.8 | 89.7 | 44.2 | 26.7 | 65.4 | 29.4 |
| Open-Sora | 47.3 | 75.9 | 22.1 | 54.8 | - | - | - | - |

Table 2: Results for action recognition on UCF101 (Soomro et al., 2012) and HMDB51 (Kuehne et al., 2011), and object discovery on MOVi-C and MOVi-E (Greff et al., 2022). The top two results are marked in green and red respectively. Stable Video diffusion stands out for the two tasks by its capacity of capturing the structure of the dynamic world.

are less effective for DiT models due to fundamental differences in how multi-modal features are fused. Developing new feature extraction methods for DiT remains an open research challenge.

**Object discovery.** We present the results for object discovery in Table 2 (right). Overall DINOv2 achieves the best performance among all the compared models, demonstrating its superior object-awareness. Among the diffusion-based models, SVD emerges as the top performer, highlighting the strong benefits of video-based training for object discovery. Interestingly, although diffusion

| Model | CityScape | | DAVIS17 | | | JHMDB | |
|---|---|---|---|---|---|---|---|
| | mIoU (SS) | mErr (Depth) | $\mathcal{J}_m$ | $\mathcal{F}_m$ | $\mathcal{J}\&F_m$ | PCK@0.1 | PCK@0.2 |
| DINOv2 | 53.6 | 4.30 | 64.8 | 69.1 | 67.0 | 50.42 | 78.71 |
| VJEPA | 41.3 | 5.27 | 52.3 | 58.0 | 55.1 | 37.55 | 70.31 |
| SD | 44.5 | 4.97 | 67.8 | 74.6 | 71.2 | 60.48 | 80.77 |
| SD3 | 46.0 | 5.09 | 48.5 | 54.8 | 51.6 | 38.17 | 65.89 |
| ModelScope | 49.3 | 3.98 | 65.3 | 72.4 | 68.4 | 60.90 | 82.83 |
| SVD | 48.1 | 4.68 | 59.8 | 67.7 | 63.8 | 60.52 | 81.84 |

Table 3: Quantitative comparisons for scene understanding on CityScape (Cordts et al., 2016) and label propagation tasks on DAVIS17 (Pont-Tuset et al., 2017) and JHMDB (Jhuang et al., 2013). The top two results are marked in green and red respectively. ModelScope achieves superior performance on the two tasks.

models fall behind in terms of FG ARI, which evaluates foreground object segmentation without considering background pixels, they outperform in terms of mBO on the MOVi-E dataset which involves more complex ego and object motion. This suggests that diffusion models are particularly effective at identifying and tracking objects in challenging motion scenarios, making them well-suited for object-centric tasks where precise localization and tracking are key. Finally, VJEPA shows weaker performance on this task, aligning with the feature visualization in Figure 1, which highlights its limitations in capturing detailed object representations.

**Scene understanding.** We report the numbers for video semantic segmentation (SS) and monocular depth estimation (DE) in Table 3 (left). ModelScope and DINOv2 merge as the top performers in these tasks, with DINOv2 excelling in semantic understanding and ModelScope showing superior performance in depth estimation. DINOv2's strong results are well-established in both segmentation and depth tasks, especially given that the evaluation on CityScape is under frame level. For ModelScope, we hypothesize that its success stems from its ability to learn semantic- and motion-aware representations, enabling it to better capture the structure of dynamic scenes. Additionally, the two video diffusion models significantly outperform image-based diffusion models in depth estimation, likely due to their capacity to leverage motion information, which inherently aids in understanding depth.

**Label propagation.** The quantitative results are shown in Table 3 (right). As we discussed above, SD3 does not work well for visual perception tasks, therefore also yielding poor results on label propagation. For the other methods, we observed that: (1) For DAVIS17, video models, both diffusion-based and non-diffusion-based ones, consistently lagged behind image-based models. The reason we hypostasize is that, video models excel in distinguishing motion (*i.e.*, moving and static) while doing relatively poorly in distinguishing between-objects and non-moving objects which are two challenging points for VOS. (2) For JHMDB, the task of pose estimation has been restricted to only focus on a single and moving object, for which video diffusion models show their strength. (3) Compared with conventional models, diffusion-based models show ability regarding semantic and temporal correspondence, making them suitable for a specific set of tasks.

**Overall conclusions.** For all of four tasks, video diffusion models consistently rank among the top performers, highlighting their robustness and adaptability in video understanding. Other key findings include that (1) Video diffusion models demonstrate exceptional proficiency in capturing motion patterns and temporal dynamics, a capability that significantly contributes to their strong performance in video understanding tasks. This unique ability opens a new avenue for advancing video understanding, offering a fresh perspective compared to traditional discriminative-based models. (2) Video diffusion models generally outperform their image-based counterparts, particularly in training-based tasks, underscoring the importance of explicitly modeling spatio-temporal information for video understanding. (3) The scale and nature of the training data play a key role in model performance. Models trained on larger datasets exhibit greater robustness, and video pretraining enhances motion modeling capabilities at the potential cost of a loss in temporal consistency when handling static objects. (4) Current diffusion-based feature extraction pipelines with UNet-based models are less effective for DiT models due to fundamental differences in how multi-modal features are fused. Developing new feature extraction methods for DiT remains an open research challenge.

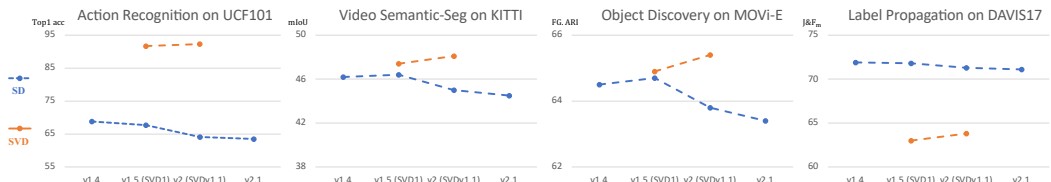

Figure 4: Model Checkpoint ablation on the four tasks. Within diffusion models, better generation capacity does not always translate to superior performance in visual perception tasks.

| Noise Level | Block Index | HMDB51 | | DAVIS17 | | |
|:---:|:---:|:---:|:---:|:---:|:---:|:---:|
| | | Top1 Acc | Top5 Acc | $\mathcal{J}_m$ | $\mathcal{F}_m$ | $\mathcal{J}\&F_m$ |
| 0 | 1 | 60.3 | 88.0 | 52.1 | 44.9 | 48.5 |
| 50 | 1 | 63.8 | **89.7** | 51.1 | 42.6 | 46.9 |
| 100 | 1 | **63.9** | 89.4 | 50.3 | 41.6 | 46.0 |
| 200 | 1 | 62.6 | 88.7 | 50.2 | 41.3 | 45.8 |
| 0 | 2 | 31.1 | 64.0 | **60.8** | **68.0** | **64.4** |
| 50 | 2 | 33.7 | 66.9 | 59.8 | 67.7 | 63.8 |
| 100 | 2 | 35.4 | 68.0 | 59.6 | 67.2 | 63.4 |
| 200 | 2 | 32.8 | 66.8 | 59.1 | 64.5 | 62.8 |

Table 4: Abalation on noise level selection and block index of SVD on HMDB51 and DAVIS17. Compared to noise level, the block index has a significant impact on downstream task performance. Features from earlier blocks capture more abstract, high-level information, while features from later blocks are more object-oriented.

## 4.3 ABLATION STUDY

**Generation v.s. perception.** A natural question arises: does a model with superior generation capacity inherently perform better in visual perception tasks? Directly comparing the generation capacity of all diffusion-based models in our evaluation is challenging, as they are designed for different tasks – some are text-conditioned, others image-conditioned, and they span both image and video generation. To address this, we conducted an ablation study using different checkpoints of the same model, based on the assumption that later versions exhibit improved generation capacity. We evaluated four versions of SD (v1.4, 1.5, 2.0, and 2.1) and two versions of SVD (v1 and v1.1) across the four tasks, with the results presented in Figure 4. Interestingly, we found that within diffusion models, better generation capacity does not always translate to superior performance in visual perception tasks. Different versions of the same model may excel in different downstream tasks, indicating that no universal metric for selecting a representation exists as of yet.

**Noisy steps and block index.** We ablate the effects of noise level selection and block index in SVD for action recognition on HMDB51 and label propagation on DAVIS17, as shown in Table 4. Our results indicate that, compared to the block index, the noise level plays a relatively smaller and more task-specific role. Pilot experiments are recommended to determine the optimal noise level when applying video diffusion models to new tasks. In contrast, the block index has a significant impact on downstream task performance. Features extracted from earlier blocks capture more abstract, high-level information, making them well-suited for classification tasks. Meanwhile, features from later blocks are more detail-oriented, which is advantageous for dense prediction tasks. This finding aligns with insights from image diffusion models, as revealed by Tang et al. (2023).

**Inference cost.** We report the inference time and memory usage for a single batch of size $[6, 256, 256]$ on the MOVi-E dataset, using an NVIDIA A100 GPU in Table 5. The baseline model, DINOv2, has an inference time of 0.224 seconds and consumes 2.6 GB of GPU memory. Notably, the memory consumption for ModelScope is an outlier, due to the lack of optimization in its public implementation. In general, diffusion-based and video-based models require more computational resources, though these costs remain acceptable. The exception is SD3, which employs a DiT-based architecture. This observation is consistent with our earlier conclusions and highlights the need for developing more efficient and effective feature extraction methods for DiT-based models.

| Model | DINOv2 | VJEPA | SD | SD3 | ModelScope | SVD |
|---|---|---|---|---|---|---|
| Memory | 1.0× | 1.1× | 1.8× | 4.6× | 8.3× | 2.7× |
| Inference time | 1.0× | 1.7× | 1.1× | 3.3× | 2.0× | 2.1× |

Table 5: Time and Memory Consumptions for all the compared models. Diffusion-based and video-based models require more computational resources but the costs remain acceptable.

| Strategy | Top1 Acc | Top5 Acc | Memory | Training time |
|---|---|---|---|---|
| Frozen | 63.8 | 89.7 | 1 × | 1× |
| UNet-ft | 68.3 | 93.5 | 8 × | 4.6 × |

Table 6: Performance and cost for finetuning video diffusion models. While finetuning the diffusion backbone yields performance improvements, it comes with significantly higher computational costs. A more efficient finetuning scheduler, such as partial finetuning with the most important parameters, is needed.

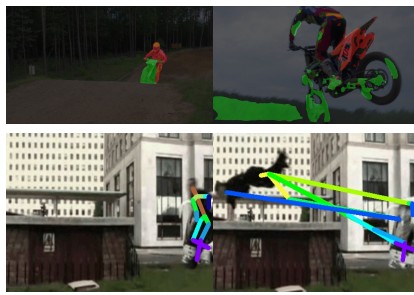

Figure 5: Failure case with large object motion for label propagation.

**Finetuning video diffusion models.** We finetuned the SVD denoising UNet on HMDB51, and the resultsm are presented in Table 6. While finetuning the diffusion backbone yields performance improvements, it comes with significantly higher computational costs. Notably, by comparing the change of parameters of all the modules, we find that the earlier downsample blocks show the highest sensitivity to parameter changes, suggesting they play a critical role in enhancing task performance. This opens up the possibility of applying partial finetuning techniques, commonly used in generative diffusion model finetuning (Kumari et al., 2023; Bao et al., 2024), to improve performance more efficiently while maintaining a lightweight training process. We consider this a promising avenue for future exploration.

**Failure cases.** We show two typical failure cases for SVD on label propagation in Fig. 5: it fails when large object motion happens, which is a general challenge for label propagation tasks. Using a specially designed parameter group or training a task head on top of the feature representation can help to solve this issue.

## 5 DISCUSSIONS

**Limitation.** In this work, we aim to analyze the strengths and limitations of video diffusion models for a variety of video understanding tasks. Our probing framework is less powerful thus the performances are still not optimal. More advanced frameworks with video diffusion models can fully explore their potential for video understanding. Moreover, though the diffusion backbones used in this paper are the open-soured ones, their performances are still far from state-of-the-art (Brooks et al., 2024). With the current rapid development of video diffusion models and the trend of open source, we believe the performance of leveraging video diffusion models for video understanding tasks can be greatly improved in the future.

Two feasible **future work** of this study include: (1) exploring suitable finetuning strategy for video diffusion models; and (2) designing a more advanced feature extraction pipeline with newly introduced DiT-based models.

**Conclusion and social impact.** This paper showcases the untapped versatility of diffusion models, encouraging the vision community to expand their application beyond generation to video understanding tasks. By pushing the boundaries of what is possible with video diffusion models, the findings in this paper can further inspire future explorations with video diffusion models in both generative and video analysis aspects.

CODE OF ETHICS

There is no obvious negative societal impact from our work. The potential negative impact is likely the same as other research leveraging large-scale generative models with the legal concern on the training data.

REPRODUCIBILITY STATEMENT

We provide extensive descriptions of the implementation details in the appendix. Also, we will release the code upon acceptance.

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

# A    IMPLEMENTATION DETAILS

## A.1    BACKBONE MODEL IMPLEMENTATIONS

We implement our backbone models including both diffusion-based ones and discriminative ones with public implementations of DIFT[1] (Tang et al., 2023), ModelScopeT2V[2] (Wang et al., 2023b), Stable Video Diffusion[3] (Blattmann et al., 2023a), Stable Diffusion 3[4] (Esser et al., 2024), DINOv2[5] (Oquab et al., 2023), and VJEPA[6] (Bardes et al., 2024b). We will release our code for reproduction upon publication.

## A.2    ACTION RECOGNITION

**Datasets.** We evaluate action recognition with top 1 and top 5 accuracy on two widely used datasets, UCF101 (Soomro et al., 2012) and HMDB51 (Kuehne et al., 2011). Both datasets are relatively small-scaled datasets containing 101 and 51 action categories respectively. UCF101 and HMDB51 contain around 9.5k/3.5k train/val videos and 3.5k/1.5k train/val videos, respectively. We center-cropped the video frames to $224 \times 224$ for evaluation and uniformly sampled 16 frames for each video for training. We build all the compared models by applying a dense layer on the averaged representation for all the video tokens to make the final prediction. Following previous works (Tong et al., 2022; Bardes et al., 2024b), we report the averaged results among the 3 test splits for both datasets.

**Training details.** We remove the data augmentation for training. Otherwise, we follow the same training procedure as VideoMAE (Tong et al., 2022). We train all the models for 100 epochs for UCF101 and 50 epochs for HMDB51. We use AdamW for optimization with a maximum learning rate of 5e-4. We use the same learning rate scheduler as the object discovery task. It takes about 1 day to train our model for UCF101 and 6 hours for HMDB51 with NVIDIA A-100 GPUS.

## A.3    OBJECT DISCOVERY

**Datasets.** We evaluate the object discovery task on two widely-used photo-realistic synthetic datasets, MOVi-C and MOVi-E (Greff et al., 2022). Both datasets feature multiple objects exhibiting rigid motion. MOVi-C solely focuses on moving objects without camera movement, whereas MOVi-E includes a mix of moving and static objects, complemented by linear camera motion. Both datasets contain 9,750 videos for training and 250 videos for validation. Each video contains 24 frames under the resolution of $256 \times 256$. We use the original resolution as the input but the mask evaluations are conducted under the resolution of $32 \times 32$, which is consistent with previous object discovery models (Kipf et al., 2022; Bao et al., 2023; Singh et al., 2022; Zadaianchuk et al., 2023; Aydemir et al., 2023). We take video foreground adjust random index (FG. ARI) and video mean best overlap (mBO) as metrics.

**Baselines.** We build our main model and the additional three variants with different pre-trained feature extraction backbones upon the public implementation of MoTok[7] (Bao et al., 2023). We replace their Resnet (He et al., 2016) feature extractor backbone with the other pre-trained feature extractors. For VJEPA, we repeat each frame twice to match the shape.

**Training details.** We build all the baseline models with 15 slots following the setting of VideoSAUR (Zadaianchuk et al., 2023). We train all the baseline models for 500 epochs with a batch size of 48. We use AdamW (Loshchilov & Hutter, 2017) for optimization with a gradient clip with norm 0.1. We apply a Cosine annealing learning rate scheduler with the largest learning rate as 5e-5 and warm-up steps as 3000. It takes about 3 days to train our model with NVIDIA A-100 GPUS.

---

[1]https://github.com/Tsingularity/dift
[2]https://github.com/ali-vilab/VGen
[3]https://github.com/Stability-AI/generative-models
[4]https://huggingface.co/stabilityai/stable-diffusion-3-medium-diffusers
[5]https://github.com/facebookresearch/dinov2
[6]https://github.com/facebookresearch/jepa
[7]https://github.com/zpbao/MoTok

### A.4 SCENE UNDERSTANDING

**Datasets.** We conduct scene understanding on CityScape (Cordts et al., 2016) datasets. We select video semantic segmentation and monocular depth estimation as the target task. CityScape dataset contains 5,000 labeled frames with a train, val, and test split. However, the labels for the test set are not of the same quality as the other two. Therefore, we evaluate on the val set, following previous work (Luo et al., 2020). The original resolution of CityScape is $1024 \times 2048$, we downsample them to $256 \times 512$ to run the evaluation. We train each model with a video clip of 16 frames.

**Training details.** We train each model for 100 epochs with a batch size of 24. We use AdamW (Loshchilov & Hutter, 2017) for optimization with a gradient clip with norm 0.1. We apply a Cosine annealing learning rate scheduler with the largest learning rate as 5e-5 and warm-up steps as 3000. It takes about 1 day to train our model with NVIDIA A-100 GPUS. We did not include any data augmentation in our training. We randomly select a video clip that contain the labeled one during training, while in inference, we start from a fixed frame where the labeled one is in the middle.

### A.5 LABEL PROPAGATION

**Datasets.** We conduct the label propagation for video object segmentation on DAVIS17 (Pont-Tuset et al., 2017) and keypoint estimation on JHMDB (Jhuang et al., 2013) following the same setup as DIFT (Tang et al., 2023). DAVIS17 is a multi-object segmentation dataset with unfixed lengths from around 40 frames to 110 frames. We evaluate our model on the resolution of $512 \times 896$ by resizing the original 480p frames. JHMDB is a keypoint estimation dataset. We follow the implementation of CRW (Jabri et al., 2020), we resize each video frame's smaller side to 320 and keep the original aspect ratio. We report region-based similarity $\mathcal{J}$ and contour-based accuracy $\mathcal{F}$ (Perazzi et al., 2016) for DAVIS17, and percentage of correct keypoints (PCK) for JHMDB.

**Hyperparameters.** For use the same evaluation pipeline as DIFT (Tang et al., 2023). The hyperparameters are listed in Table A. We cite the results for all the other methods from DIFT (Tang et al., 2023).

| Dataset | Time step $t$ | Block index $n$ | Temperature for softmax | Propagation radius | $k$ for top-$k$ | Number of prev. frames |
|---|---|---|---|---|---|---|
| DAVIS-2017 | 25 | 2 | 0.1 | 10 | 15 | 28 |
| JHMDB | 25 | 2 | 0.1 | 5 | 15 | 14 |

Table A: Hyperparatemers for the label propagation tasks.

## B ADDITIONAL VISUALIZATIONS

We show additional visualizations with backbone SVD on object discovery, and ModelScope on label propagation on Figs. A, B, respectively, showing the promising results of video diffusion features for video understanding.

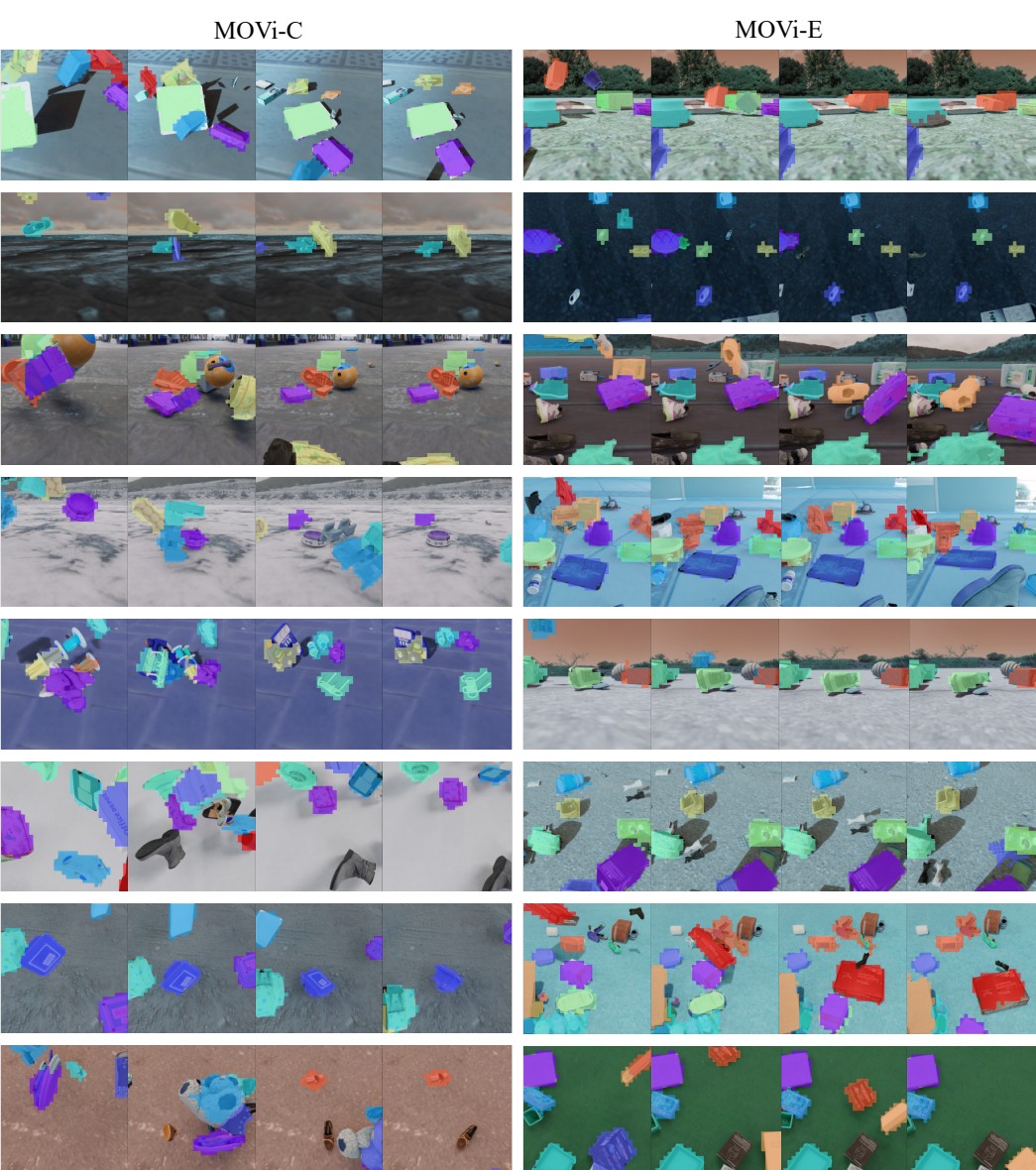

Figure A: Additional visualizations for object discovery on MOVi-C and MOVi-E datasets (Greff et al., 2022) with SVD (Blattmann et al., 2023a) as backbone. We show the Top 10 object masks for each method and ignore the background masks for better visualizations. Our model achieves promising results for object discovery tasks.

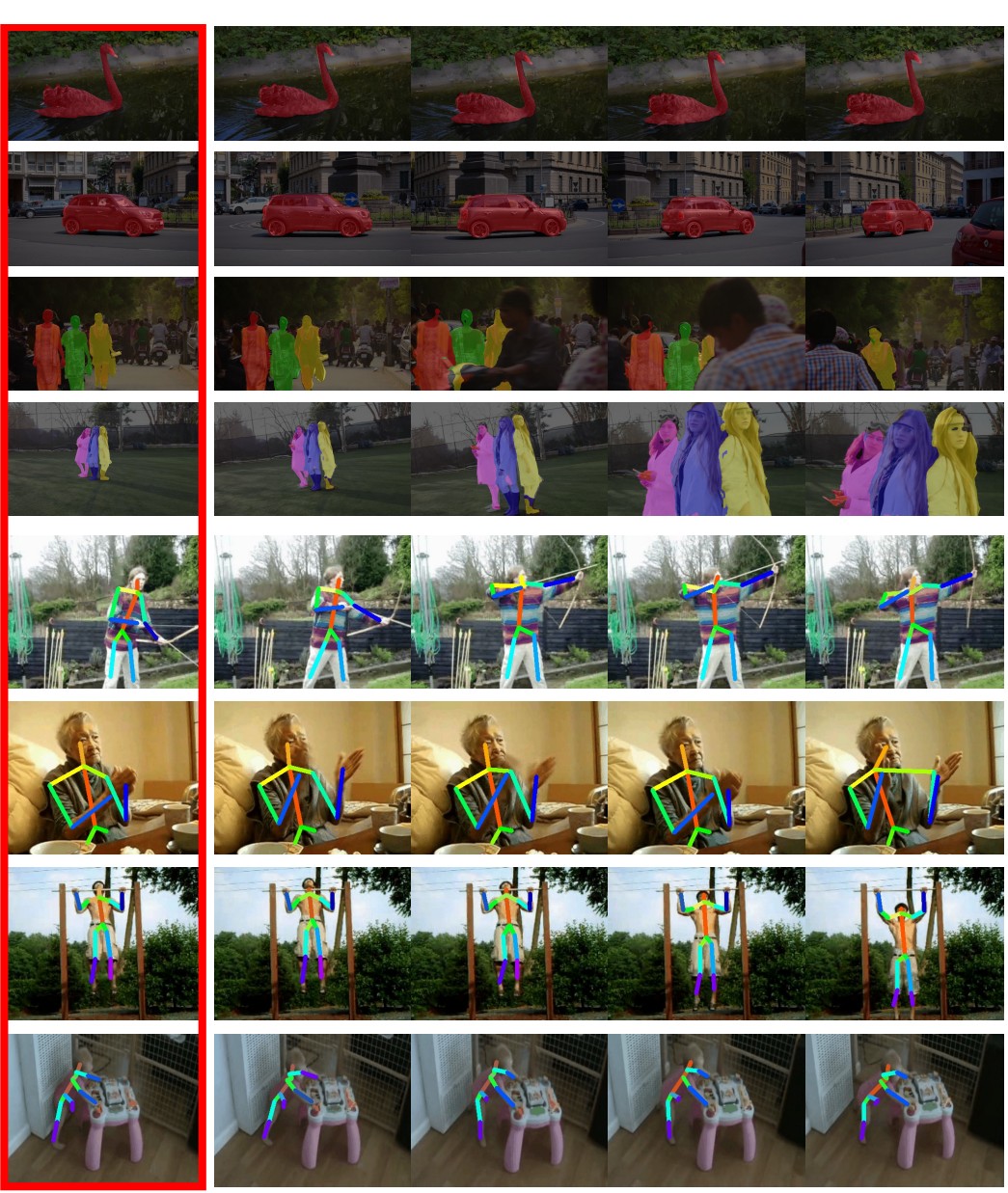

Figure B: Additional visualizations for label propagation tasks with ModelScope (Wang et al., 2023b) as the backbone. Top 4 rows: video object segmentation task on DAVIS17 (Pont-Tuset et al., 2017); Bottom 4 rows: keypoint estimation on JHMDB (Jhuang et al., 2013).

