# OpenReview forum: "Video Diffusion Models Learn the Structure of the Dynamic World"
_ICLR.cc/2025/Conference — ICLR 2025 Conference Withdrawn Submission_

### Official Review · Reviewer_cQqH · 2024-10-29

**Soundness:** 2
**Presentation:** 2
**Contribution:** 2
**Rating:** 5
**Confidence:** 5

**Summary:**

This paper explores the adaptation of video diffusion models to fine-grained video understanding tasks, such as action recognition and object discovery. The authors conducted comparative experiments on several different video diffusion models and found that the spatiotemporal modeling of generative models can be effectively applied to video understanding tasks.

**Strengths:**

- The motivation and writing of this paper are very clear and easy to follow.
- The experiments are quite rich, comparing various feature extractors and multiple different tasks.

**Weaknesses:**

- The novelty is somewhat limited, as the method in this paper is inspired by DIFT from the image domain. The extraction of visual features also follows its diffusion feature extraction approach.
- The method section lacks insights, mainly introducing and simply applying others' methods.
- Regarding the features and training methods, most of the approaches in this paper are fixed, i.e., they freeze the backbone and only tune the task head. Exploring features from different layers and fine-tuning more backbone parameters could be valuable and might improve the model's performance.
- Although this paper conducted experiments on multiple tasks, it did not achieve good results for each individual task. This limits the practical applications and impact of the paper.
- There is a lack of citations for some key literature, such as GenRec[1], which also explores the application of video diffusion models in action recognition and examines the performance on K400 and SSv2. In contrast, this paper only validates its results on the smaller datasets UCF and HMDB, which are not representative.

Given these issues, I can only give a borderline rejection score for now, and I look forward to the authors' response.
[1] Weng, Zejia, et al. "Genrec: Unifying video generation and recognition with diffusion models." NeurIPS 2024.↳

**Questions:**

See above in weakness .

---

> ### Author Response · Authors · 2024-11-15
> **Response to reviewer cQqH**
>
> Thank you for your detailed review and feedback. We appreciate your insights and have addressed each of your points below, outlining our planned revisions.
>
> Q: **Novelty with respect to DIFT**
> A: We understand your concern regarding the limited novelty of our work, given its inspiration from DIFT. While our feature extraction approach is indeed rooted in prior work, our study extends DIFT’s application to a wide variety of video understanding tasks, providing new insights into how video diffusion models balance spatiotemporal modeling for downstream tasks. From the tech perspective, we also differ from DIFT with (1) we apply trainable task heads for a variety of video understanding tasks while they purely study the semantic correspondence in the feature level and (2) We also study feature extraction pipelines with diffusion-transformers, which is different from UNet-based architectures with details provided in the appendix.
>
> Q: **Insights in Method Section**
> A: While we apply existing methods in our study, our contribution lies in systematically analyzing these models in the video domain and uncovering actionable insights. For example, we found that video diffusion models' representations vary by layer, with earlier layers capturing global information while later layers focus on detailed, object-centric features. This distinction provides a useful guideline for targeted fine-tuning, where only specific layers might need adaptation for particular tasks, making the process more efficient. We will highlight these insights more explicitly in our revision. Moreover, we demonstrate that video diffusion models offer a unique combination of object motion and scene structure capabilities, which makes them valuable in tasks requiring both global and local understanding of video content. Additionally, we observe that while diffusion models excel in generation, this does not always translate into downstream performance—a novel finding that we believe adds depth to the field's understanding of generative models in video.
>
> Q: **Limited Fine-Tuning of Backbone Parameters**
> A: We agree that exploring feature representations from different layers and allowing more backbone parameters to be fine-tuned could potentially improve performance. However, extensive fine-tuning of large diffusion models across a variety of tasks is computationally intensive and would be beyond the current scope of this study. To address this, we propose targeted layer-wise fine-tuning as an efficient approach, based on our findings that different layers capture information at different granularities. This approach could balance computational efficiency with performance improvements, a point we will clarify in our revision.
>
> Q: **Performance on Individual Tasks**
> A:  We recognize that video diffusion models may not outperform specialized models on every individual task. However, our results highlight their versatility, as they achieve competitive performance across a wide range of tasks, making them suitable for settings where multiple tasks must be addressed without switching models. We will further discuss this trade-off in the paper, highlighting the broader applicability of video diffusion models while acknowledging their limitations in specialized scenarios.
>
> Q: **Literature Citations and More Challenging Datasets**
> A: Thank you for bringing GenRec to our attention. We will include a discussion and citation of GenRec in our paper and consider how its findings relate to our own. Additionally, we agree that more challenging datasets such as Kinetics-400 and Something-Something v2 would better demonstrate the temporal dynamics captured by video diffusion models. We will conduct additional experiments on these datasets to strengthen our study and include the results in our revised manuscript.

---

### Official Review · Reviewer_uAKH · 2024-11-01

**Soundness:** 3
**Presentation:** 3
**Contribution:** 2
**Rating:** 5
**Confidence:** 4

**Summary:**

This paper focuses on analyzing the strengths and limitations of existing video diffusion models for different video understanding tasks. A probing framework is proposed to extract the video feature representations from existing pre-trained models and learn lightweight task heads for different downstream video understanding tasks.  Comprehensive experiments across six models and four downstream video tasks are conducted and some findings are demonstrated.

**Strengths:**

1) The investigation of different characteristics of image-/video-based diffusions is interesting and meaningful for the video-understanding community;
2) The proposed probing framework is technically sound and the experimental results are comprehensive.
3) The paper is well-written, making it easy to follow and understand.

**Weaknesses:**

1) Although many findings are demonstrated by the comprehensive results, **the reviewer finds these findings are well-known consensus in the video understanding community**. For example, modeling the human motion or object dynamics is important for analyzing the videos, it is straightforward that most video-based diffusions contribute their models to this problem. Therefore, for instance, the findings "video diffusion models demonstrate exceptional proficiency in capturing motion patterns and temporal dynamics" that the paper try to demonstrate, **cannot provide new insights to the community**;

2) The used video action recognition datasets, i.e., UCF and HMDB, actually do not require modeling too much temporal/dynamic semantics for recognition. To make the demonstration more convincing, the reviewer thinks experiments should be conducted on the action datasets that require effective temporal dynamics modeling (e.g., something v2, Kinetics).

minor issues:
1) the symbol zT in line #231 should be corrected;
2) the sentence in line #289~290 should be revised;

**Questions:**

please see the weaknesses section.

---

> ### Author Response · Authors · 2024-11-15
> **Response to reviewer uAKH**
>
> Thank you for your constructive feedback and for recognizing the relevance of analyzing diffusion models for video understanding tasks. We appreciate your insights and have outlined our responses and planned adjustments below.
>
> Q: **Novelty of Findings**
> A: We understand the concern that some findings, such as the importance of modeling motion and object dynamics, may appear to reinforce existing knowledge within the video understanding community. However, our empirical study offers new insights into how video diffusion models capture these elements, as well as the specific trade-offs they present. In particular, our findings show that video diffusion models uniquely balance capturing object motion and scene structures, offering a distinctive combination that we believe has yet to be fully demonstrated in the community.
>
> Furthermore, our probing framework revealed a notable layer-specific characteristic: earlier layers in video diffusion models retain global, scene-level information, which supports video-level tasks, whereas later layers offer more detailed, object-centric features. This observation provides actionable insights for model fine-tuning, suggesting that focusing on layer-specific adaptation can yield an efficient training process while maintaining strong performance. Additionally, we found that superior generative abilities do not necessarily correlate with improved downstream task performance, underscoring an important distinction for future work in optimizing video diffusion models for understanding tasks specifically.
> We will emphasize these nuanced findings in our revision, as they underscore the empirical contributions of our work and provide fresh perspectives on using diffusion models for video understanding tasks.
>
> Q: **Evaluation on Action Recognition Datasets**
> A: Thank you for this suggestion. We agree that including action recognition datasets with greater emphasis on temporal dynamics, such as Something-Something v2 and Kinetics, would strengthen our study. We plan to conduct additional experiments on these datasets to further validate our findings and provide a more robust assessment of video diffusion models’ capabilities for modeling temporal dependencies. The results from these datasets will be included in our revised manuscript to enhance the rigor and relevance of our evaluation.

---

### Official Review · Reviewer_qjWk · 2024-11-02

**Soundness:** 2
**Presentation:** 2
**Contribution:** 2
**Rating:** 3
**Confidence:** 5

**Summary:**

This paper analyzed the capability of video diffusion model in video understanding tasks. They extracted features from diffusion model, and used these features to perform several video tasks, including action recognition, object discovery, scene understanding, and label propagation. Results show that video diffusion can achieve good results.

**Strengths:**

-The analyze of video diffusion models on understanding tasks is important.

**Weaknesses:**

-While the aim of using diffusion model in understanding tasks is appreciated, the designed method is not new compared to diffusion features (DIFT). This work is more like a technical report, and provides little insights.

-From the results on four tasks, I don't think video diffusion models have significant advantages compared to SSL models and image diffusion models, considering that they have been trained on vast videos. In action recognition, SVD actually performs similar to VJEPA. In object discovery, DINOv2 performs much better. In scene segmentation, DINOv2 and ModelScope are the best on two metrics. In label propagation, SVD/ModelScope performs similar to SD. Particularly, since SVD is extended from SD 2.1, while the performance gap between SD and SVD on object discovery, scene understanding, and label propagation is quite minor, considering the enormous training cost of SVD, I don't think it is promising.

--Minor, typo in line 232, zT, \epsilon^v\theta. Line 290, MOT17(), report xxx. Many depulicated sentences, e.g. Line 372-373 and Line 430-431.

**Questions:**

See the weaknesses.

---

> ### Author Response · Authors · 2024-11-15
> **Response to reviewer qjWk**
>
> Thank you for your thoughtful review and for recognizing the importance of analyzing video diffusion models in understanding tasks. We appreciate your feedback and have addressed each of your points below.
>
>
>
> Q: **Novelty and Insights Beyond DIFT**
> A: We understand your concern regarding novelty relative to prior work like DIFT. Our study is designed as an empirical exploration, which we believe offers valuable contributions through its systematic analysis of video diffusion models across various video understanding tasks. While we do not propose a new model, our key insights include the following:
>
> - Balance of Object Motion and Scene Structure: Video diffusion models demonstrate a unique balance in capturing object motion and scene structure, which is essential for video understanding tasks.
> - Layer-Specific Representations: We observed that different layers in video diffusion models carry information at varying levels of granularity. Earlier layers retain more global, scene-level information, suitable for tasks like action recognition, while later layers offer more detailed object-level features. This distinction informs our suggestion for targeted fine-tuning, where selectively tuning these important layers could yield an efficient training process while maintaining strong performance.
> - Generation vs. Downstream Performance: Interestingly, we found that better generation capabilities do not necessarily correlate with superior downstream task performance. This finding highlights a potential area for future work in optimizing video diffusion models specifically for understanding tasks rather than generation alone.
>
>
> We will emphasize these insights more explicitly in our revision, underscoring the contributions of our empirical study as a research paper rather than a technical report.
>
> Q: **Comparative Performance and Training Cost**
> A: We appreciate your observations regarding the comparative performance of video diffusion models versus SSL models and image diffusion models. While it is true that video diffusion models may not outperform SSL models on all individual tasks, they provide the most balanced and consistent performance across the diverse set of video understanding tasks we evaluated. This consistent, versatile performance is one of the strengths of video diffusion models, especially in settings where model-switching for task specialization is impractical.
>
> Regarding training costs, we acknowledge the high computational demands of training video diffusion models like SVD. However, the incremental cost is not substantial when using a pre-trained Stable Diffusion (SD) as initialization, which is a standard practice in the literature.
>
> We will further clarify these points in our paper to contextualize the advantages and limitations of video diffusion models alongside the cost implications, making it easier for readers to assess the trade-offs involved.

---

### Official Review · Reviewer_ttdC · 2024-11-02

**Soundness:** 2
**Presentation:** 2
**Contribution:** 2
**Rating:** 3
**Confidence:** 4

**Summary:**

The proposed paper aims to evaluate video diffusion models with respect to their visual perception capabilities. To this end, it analyzes the feature representations learned by image- and video-based diffusion models in context of video understanding tasks like action recognition, object detection, scene understanding, and label propagation on datasets like UCF101, HMDB, MOVi-C, MOVi-E MOT, DAVIS, and JHMDB.

**Strengths:**

The topic if and how diffusion models are able to capture visual representations is of great interest. The topic is also very timely.

**Weaknesses:**

- 1. Comparison to SoTA/RW:
Compared to the original DIFT paper, results in Tab 3. are below state-of-the-art and varying also with respect to DIFT baselines. Namely, DIFT (from Stable Diffusion) reports e.g. PCK@0.1=61.1 for JHMDB (PCK@0.1=60.48 for SD and PCK@0.1=60.52 for SDV in the paper), deviation for DAVIS is even worse.

- 2. Evaluation of video diffusion vs general visual foundation models?:
The evaluation considers three Video diffusion models(SVD, ModelScope, Open-Sora), and four other models (e.g. image diffusion, Dinov2, VJEPA). This is a bit irritating and not clarified in the evaluation table or the discussion. It would be better to account for those scenarios as well in the paper (e.g. by a comparative discussion of different representations based on different pretrainings etc.)
Further, if the paper would care about video in the first place, why not test against more video models? While VJEPA is one possible backbone, it would be good to have a least a few more representatives like VideoMAEv1 or v2, InternVid, etc.

- 3. Details about video probing:
The video probing is very superficially described (imho). As this is the most novel part of the paper, spending only 20 lines on this makes it challenging to understand what's really going on. I really tried to consider if the conclusion in the experiments make sense to me, but I always ended up with having the feeling that details are missing. Some were available in the experiments later, but it was hard to get a full picture. The paper should be more self-contained.

- 4. Feature computation (line 299-303):
It seems like the block index is chosen per model (which makes sense), but it is unclear based on which criteria. Was this optimized per model per task as shown in Tab. 4?

**Questions:**

There are some typos, e.g. line 290 "... and report xxx."

While I don't think that this would be a feasible improvement to do for this specific conference (bc it's mainly a new paper), some suggestions for a revision could be (adapted from what the OpenReview LLM proposed based on the weakness section, but nice anyway):

- 1. Give a detailed background section about the original DIFT paper and directly address these discrepancies  (if there are any). Provide a detailed comparison and explanation for why your results differ, and discuss any implications this may have on your conclusions.

- 2. Clearly categorize the models in their evaluation tables (e.g., video diffusion models vs. other types) and discuss and provide a rationale for including non-video models in the analysis.

- 3. Explain your criteria for selecting the models included in the study, and consider expanding the analysis to include additional, prominent video models.

- 4. Provide a more comprehensive description of the video probing methodology in the main body of the paper. Include specific details on techniques used, implementation methods, and key parameters or settings.

---

> ### Author Response · Authors · 2024-11-15
> **Response to reviewer ttdC**
>
> Thank you for your valuable feedback on our work. We appreciate your insights and have outlined below how we plan to address each of the points raised.
>
> Q: **Comparison to DIFT**
> A: Please note that the DIFT label propagation framework, which we adopted for these experiments, was highly tuned to the StableDiffusion features used by DIFT’s authors. Their improved results stem from a non-standard StableDiffusion variant (ADM version) trained on ImageNet, which achieves higher performance on this benchmark. This highlights the specificities of optimizing for DAVIS and achieving state-of-the-art results on this small dataset.
>
>
> Q: **Evaluation Scope and Additional Video Models**
> A: We appreciate your suggestion to broaden the scope of our evaluation by including more video-centric models. To address this, we will add additional video representations such as VideoMAEv1, VideoMAEv2, and InternVid to our experiments. We will also improve the presentation of results to explicitly compare different pretraining schemes (image-based versus video-based diffusion and other foundation models) and discuss their respective strengths and limitations. This comparison will provide a clearer understanding of the representational differences across model types.
>
> Q: **Details on Video Probing Framework**
> A: We recognize the need to make the linear probing framework more comprehensible and self-contained. To improve clarity, we will expand the description of the video probing setup, outlining specific choices and decisions behind each step. This expansion will provide the full context behind the probing experiments, allowing readers to follow our methodology more transparently and fully understand how we reach our conclusions.
>
> Q: **Feature Computation and Block Index Selection**
> A: Thank you for pointing out the need for further detail here. The block index selection is not optimized per task per model. We follow the rule (Inspired by DIFT and verified in Table 4) that uses an earlier block index for global understanding tasks and a later block index for dense pixel prediction. We use a relatively small and fixed noise level for all the models and all the tasks. For DiT-based models using ratified flow, we recomputed the noise level for them to match the actual timestep used for other models. We will further clarify this selection in our future revision.

---

### Note · Authors · 2024-11-15

I have read and agree with the venue's withdrawal policy on behalf of myself and my co-authors.